# Magnetic Polymers for Magnetophoretic Separation in Microfluidic Devices

**Lucie Descamps** [1,2], **Damien Le Roy** [2], **Caterina Tomba** [1] **and Anne-laure Deman** [1,*]

1   Univ Lyon, Université Claude Bernard Lyon 1, CNRS, INSA Lyon, Ecole Centrale de Lyon, CPE Lyon, INL, UMR5270, 69622 Villeurbanne, France; Lucie.descamps@univ-lyon1.fr (L.D.); caterina.tomba@univ-lyon1.fr (C.T.)
2   Institut Lumière Matière ILM-UMR 5306, CNRS, Université Lyon 1, 69622 Villeurbanne, France; damien.le-roy@univ-lyon1.fr
*   Correspondence: anne-laure.deman-him@univ-lyon1.fr

**Abstract:** Magnetophoresis offers many advantages for manipulating magnetic targets in microsystems. The integration of micro-flux concentrators and micro-magnets allows achieving large field gradients and therefore large reachable magnetic forces. However, the associated fabrication techniques are often complex and costly, and besides, they put specific constraints on the geometries. Magnetic composite polymers provide a promising alternative in terms of simplicity and fabrication costs, and they open new perspectives for the microstructuring, design, and integration of magnetic functions. In this review, we propose a state of the art of research works implementing magnetic polymers to trap or sort magnetic micro-beads or magnetically labeled cells in microfluidic devices.

**Keywords:** magnetic polymers; polymer composites; magnetophoresis; micro-bead separation; cell separation; microfluidic devices

## 1. Introduction

Microfluidics inspired vivid interest in biomedical applications as it meets a need for the manipulation of micro- and nanoscale objects by offering appealing features. Among them, we can cite: (i) its micrometric dimensions and laminar flow nature, enabling precise object manipulation and single-cell study; (ii) the handling of small quantities of volume, which facilitates the analysis of rare or expensive samples and speeds up processes, leading to cost-effective devices; (iii) the integration of various functions (mixing, focusing, sorting, trapping, detection, etc.) into a single device, leading to compact and portable systems and therefore opening the way for the implementation of point-of-care devices.

The manipulation of micro- and nano-objects requires external forces, such as acoustic, electrical, optical, or thermal actuations. In particular, magnetic forces are suitable for this purpose. Magnetic force-based manipulation relies on magnetophoresis, which refers to the motion of magnetic particles or magnetically labeled cells when subjected to a non-uniform magnetic field. Magnetophoresis [1–5] has been demonstrated as an efficient way to trap and separate biological entities, be it DNA [6–8], proteins [9–11], beads [12], or cells [13–16]. This strategy benefits from several advantages compared to its alternatives: (i) the contactless manipulation, which makes this technique nondestructive for biological samples and preserves cell viability/integrity; (ii) the specificity, since magnetic fields increase the magnetic contrast of non-magnetic objects, either using magnetic labels [17] or with the aid of magnetic fluids [18], (iii) the low sensitivity to medium parameters, such as surface charges, ionic concentration, pH, and temperature; and (iv) the tunability, as the magnetic force depends on the particle size, the magnetic properties of the target and surrounding medium, as well as the gradient of the magnetic field. The magnetic force can be attractive, i.e., oriented along the magnetic field gradient (positive magnetophoresis), or repulsive, i.e., oriented in the direction opposite to the magnetic field gradient (negative

magnetophoresis), depending on the apparent magnetic susceptibility of the target particle in its medium. Positive magnetophoresis is the most widespread manipulation method and occurs when magnetic objects (particles or labeled cells) are suspended in a diamagnetic fluid. On the contrary, negative magnetophoresis is a label-free technique, where diamagnetic objects are suspended in a magnetic fluid (paramagnetic salt solution or ferrofluid). However, the lesser visualization and viability of cells in ferrofluids can be a limitation to the widespread use of this approach in biomedical applications [5]. Both methods rely on the generation of high magnetic field gradients that can be controlled by various types of magnetic field sources. It should be mentioned that in the case of highly concentrated nanoparticle solutions, collective effects can contribute to an increase in the magnetic forces involved [19,20]. In the general case of separating any type of bead or cell solution, many researchers have rather focused on optimizing sources of high magnetic field gradients. The easiest approach consists in placing a macroscale permanent magnet in the vicinity of the microfluidic channel in which magnetic objects are flowing. NdFeB is the material of choice for macro-scale magnets, as it offers the highest values of remanent induction (Br) at room temperature among other hard magnetic materials. However, this approach suffers from the large distance between the macro-magnet and the microchannel, which limits the strength of the magnetic field gradient applied on the flowing magnetic objects. In addition, scaling down the size of magnetic field source scales up the magnetic field gradients. Therefore, it is beneficial to integrate micro-scale magnetic sources in microfluidic systems. Three main approaches are used to generate localized micro-magnetic field gradients: current carrying micro-coils, micro-concentrators made of soft ferromagnets (mainly Ni and Fe-Ni alloys) magnetized by an external magnetic field, and permanently magnetized micro-magnets, made of hard ferromagnetic materials (usually NdFeB). Despite the flexibility in controlling the intensity of the magnetic field when using micro-coils, Joule heating limits the magnetic field to a few tens of mT when operating in static conditions [21]. In contrast, micro-concentrators and micro-magnets can produce relatively strong magnetic fields (of a fraction of a Tesla) and thus are particularly well suited for these applications. Nevertheless, challenges remain regarding the complexity of microfabrication of magnetic micro-sources and their integration with polymer-based microfluidic systems. Film-based approaches, in which magnetic elements are integrated through standard microfabrication techniques such as sputtering [22–24], electroplating [25–29], thermal deposition [30,31], or thermo-magnetic patterning [32,33], led to unrivalled control over the reproducibility, shape, and microstructuration. However, these approaches suffer from poor adhesion with polymer substrates, the difficulty of achieving large aspect ratio microstructures, and the need for expensive and tedious fabrication processes. Other strategies have been explored to implement microstructured magnetic sources in microfluidic devices: the introduction of ferromagnetic wires (Ni, Fe-Ni) in microchannel [15,34], the coating of 3D hot-embossed thermoplastic microstructures with a thin layer of nickel [35], or the use of magnetic polymers. Magnetic polymers are composite polymers obtained by the powder-based approach, i.e., by doping the polymer matrix with magnetic particles or filaments.

Composite polymers have recently emerged as a real breakthrough for the compatible and cost-effective integration of magnetic materials into polymer-based MEMS and microfluidic devices [36,37]. In general, the composite approach allows conferring new properties to the polymers and finds many applications in the field of smart devices [38]. Concerning magnetic composite polymers dedicated to microfluidic systems, this approach enables the tailoring of the magnetic function depending on the nature, the size, the concentration, and the morphology of the magnetic powder, the nature of the polymer matrix, and the microfabrication method. Various polymer materials have been investigated for microfluidic applications: elastomers such as polydimethylsiloxane (PDMS) [39], photosensitive resists such as SU-8 [40], or thermoplastics such as polymethylmethacrylate (PMMA) [41]. Magnetic PDMS is the most commonly encountered due to the microfabrication properties of PDMS by soft lithography and the massive use of the latter for the realization of microfluidic systems. Thus, a large panel of microfluidic functionalities for

fluid sample handling has emerged employing composite polymers, such as micro-valves, micro-pumps, or micro-mixers for microfluidic flow control [36,40,42–47]; dynamic artificial cilia [41,48–50]; and reversible microchannel bonding [51]. Ferrofluids were also explored for actuation in microfluidic systems [52–55].

Magnetic polymers have also been used in microsystems to manipulate magnetic entities such as labeled cells or magnetic micro-beads by magnetophoresis. Several reviews published in the last few years show the richness of the literature and the vitality of magnetophoretic devices [56,57]. They also reveal that in their large majority, works are based on classical techniques of microelectronic manufacturing. In this review, we focus on recent works using composite magnetic polymers to sort or trap magnetic targets by magnetophoresis in microfluidic devices. We first present implementations based on PDMS composites, and in a second part, we present the other methods, which are based on magnetic fluids. Table 1 summarizes examples of magnetic composites in microsystems for sorting or trapping applications, which will be further described in the following sections.

**Table 1.** Examples of magnetic composites in microfluidic devices for magnetophoretic applications.

| Host Material | Doping Agent | | | Application | Implementation | Reference |
|---|---|---|---|---|---|---|
| | Nature | Particle Diameter | Concentration | | | |
| PDMS | Carbonyl iron | 7 μm (Sigma-Aldrich) | 50–83 wt % | Micro-bead sorting and cell trapping | Pillars inside the channel | [58] |
| PDMS | Nickel | 50 nm (DeKeDaoKing, Beijing, China) | N/A | Magnetic bead and cell trapping | Pillars inside the channel | [25] |
| PDMS | Carbonyl iron | 1–3 μm (HQ grade, BASF, Germany) | N/A | Nano-bead trapping | Pillar inside the channel | [59] |
| PDMS | Neody-mium oxide | 5 μm (Molycorp.) | N/A | Immuno-magnetic sorting of beads | Pillars inside the channel | [60] |
| PDMS | Carbonyl iron | N/A | N/A | Magnetic bead conveyor belt | Mushroom-shaped structures buried under the channel | [61] |
| PDMS | Carbonyl iron | N/A (Sigma-Aldrich) | 75 wt % | Cell trapping and sorting | Composites stripes under the channel | [62] |
| PDMS | NdFeB | 5 μm (MQFP-B, Magnequench) | 66 wt % | Cell trapping and sorting | Composites stripes under the channel | [62] |
| PDMS | NdFeB | N/A | N/A | Microfluidic mixer | Composites stripes under the channel | [63] |
| PDMS | $Fe_3O_4$ | 50–100 nm (637106, Sigma-Aldrich) | 38 wt % | Trapping of magnetically labeled Vorticella | Composite blocks in the channel walls | [64] |
| PDMS | Iron | 1–6 μm (GoodFellow) | 44, 60, 70 wt % | Extraction and redispersion of functionalized magnetic particles | Integrated magnetic structure in the channel wall | [65] |
| PDMS | Carbonyl iron | N/A (C3518, Sigma-Aldrich) | 50, 66.7 wt % | Magnetic particle separation | Microstructured composite next to the channel | [66] |
| PDMS | NdFeB | N/A (MQFP-B-20076, Magnequench) | 66.7 wt % | Magnetic particle separation, Microfluidic mixer | Microstructured composite next to the channel | [67] |
| PDMS | Carbonyl iron | 7 μm (Sigma-Aldrich) | 83 wt % | Micro-bead trapping and magnetic force measurement | Self-ordered composite block in the channel wall | [46] |
| PDMS | NdFeB | N/A (MQFP-B-20076, Magnequench) | N/A | Cell sorting | Self-ordered composite under the channel | [68] |
| PDMS | Carbonyl iron | 0.5–7 μm (Sigma-Aldrich) | 1–5 wt % | Micro-bead trapping | Columnar agglomerates under the channel | [69,70] |

**Table 1.** *Cont.*

| Host Material | Doping Agent | | | Application | Implementation | Reference |
|---|---|---|---|---|---|---|
| | **Nature** | **Particle Diameter** | **Concentration** | | | |
| PDMS | NdFeB | 0.5–7 µm (MQFP-B, Magnequench) | 1 wt % | Micro-bead trapping | Columnar agglomerates under the channel | [71,72] |
| Light Hydro-carbon Oil | Magnetite | 10 nm (EMG900, Ferrotec) | N/A | Cell sorting | Microchannel parallel to the sorting channel | [73] |
| Carbon ink | Iron | 10 µm | 25 wt % | Bead trapping | Magnetic tracks perpendicular to the sorting channel | [74] |
| Water [1] | Cobalt ferrite | N/A (MJ300, Liquid Research) | N/A | Cell trapping | Integrated magnetic structures in the channel wall | [75] |
| 50% Ethanol [2] | Iron | 40 µm | N/A | Cell sorting | Channels on the side of the sorting channel | [76] |
| Water [2] | $Fe_3O_4$ | N/A | 0.2 m/v% | Particle sorting | Magnetic pole arrays close to the sorting channel | [77] |
| Water [1] | Cobalt ferrite | N/A (MJ300, Liquid Research) | ≈10 wt % | Cell trapping | Spot arrays at the bottom of the channel | [78] |

[1] Water is evaporated after heating. [2] Liquid host is removed after filtering.

## 2. The PDMS Composite Approach

PDMS composites are excellent candidates for the integration of active functions into PDMS microsystems. There are many examples in the literature of dielectrophoretic functions based on conductive PDMS [79,80] and magnetic functions based on magnetic PDMS. Magnetic PDMS composites are mainly obtained by mixing soft (Fe, Ni, and Ni-Fe alloys) or hard (NdFeB, ferrites) magnetic powders with a PDMS mixture (base polymer and curing agent). By modifying the nature, shape, concentration, and organization of the doping particles, it is possible to modulate the magnetic properties of the composite materials. One of the major advantages of these composites is that they preserve some properties of PDMS such as micropatterning by soft lithography and surface activation by $O_2$ for plasma bonding with glass and PDMS substrates. It also allows obtaining magnetic microstructures of several micrometers in thickness and with aspect ratios that are hardly obtained with conventional microfabrication techniques. In addition, the composite microstructure can be directly integrated into the microchannels, in a one-step soft-lithography process, avoiding tedious alignment procedures. This very versatile approach allows localizing the magnetic structures inside the channel or in its close vicinity, underneath or on the sides. Moreover, as the magnetic structures are integrated into PDMS microsystems, the polymer matrix being the same for the whole system, the magnetic function is tightly integrated and does not raise heterogeneous integration issues.

### 2.1. High Concentrated PDMS Composites

In 2014, two groups published research on the integration of magnetic PDMS pillars inside microchannels to trap magnetic targets. Faivre et al. obtained composite pillars by first mixing carbonyl iron particles (7 µm) and PDMS (I-PDMS) until obtaining a homogeneous material before polymer reticulation [58]. Different iron concentrations ranging from 50 wt % to 83 wt % were tested and molded in SU-8 molds of various sizes and shapes. Finally, through a one-step soft-lithography process, they integrated into microchannels I-PDMS pillars of diamond shape @83 wt %, 500 µm in diagonal, and of the same height as the channel (about 40 µm). Figure 1a shows the trapping of superparamagnetic beads on the composite pillars in the presence of external magnets and their release after magnet removal. They demonstrated the trapping of magnetic

micro-beads, magnetically labeled cells, and the separation of superparamagnetic and diamagnetic micro-beads at flow rates ranging from 50 to 200 µL/h. Yu et al. realized arrays of Ni-PDMS pillars of different shapes of about 40 µm in height and 50 to 100 µm in size [25]. They did not mix the magnetic powder with the PDMS beforehand but obtained these pillars by directly filling the SU-8 molds with nickel powder (50 nm) and then casting the PDMS on the molds. They trapped fluorescent magnetic beads and magnetically labeled yeast cells and obtained bead and cell patterns. In a similar way, Ezzaier et al. filled the SU-8 mold with carbonyl iron microbeads (1–3 µm in diameter), which were then covered by PDMS [59]. Thus, they fabricated a magnetizable micro-pillar (diameter and height of 50 µm) in a microfluidic channel to study the magnetic separation of iron oxide nanoparticles (IONPs) for immunoassays. In the presence of an external magnetic field, IONPs (27 nm in diameter) accumulated around the micropillar along the direction of the flow and of the magnetic field, with a comet-like shape increasing over time. More recently, Bae et al. integrated pillar-like permanent micromagnets made from a mixture of a hard ferromagnetic powder, neodymium oxide (5 µm nominal diameter), and PDMS [60]. The magnetic powder and the PDMS were mixed before filling the molds. The molds of the circular pillars were realized with silanized PDMS. Nd-PDMS pillars of diameter ranging from 100 to 300 µm, and presenting a height of 240 µm, were obtained (Figure 1b). To convert the pillars into permanent magnets, they were magnetized with a magnetic field of 2.5 T. They implemented the array of pillars for immuno-magnetic separation using a streptavidin–biotin binding model of polymeric micro-beads and magnetic nanoparticles. Polymer micro-beads, 1 µm in diameter, were coated with streptavidin and magnetic nanoparticles (200 nm in diameter) with biotin. Capture efficiency of 94.9% was measured at 20 µL/min.

Other groups integrated magnetic PDMS structures under the microfluidic channel or on its sides. Thus, Pelt et al. realized microfluidic magnetic beads conveyor belts [61]. Their actuation method is based on the use of mushroom-shaped soft-magnetic structures located under the channel associated with an external rotating magnetic field. They were realized by mixing an abundance of carbonyl iron particles with pre-mixed PDMS and pouring it on the PDMS molds. Mushroom-shaped microstructures were buried at 20 µm under the microfluidic channel. They injected superparamagnetic beads (2.8 µm in diameter) and observed that they formed agglomerates that rolled over the surface of the channel, just above the magnetic structures. When applying field rotation frequencies between 0.1 and 50 Hz, the beads moved at speeds greater than 1 mm/s for the highest frequency. In another example, Royet et al. fabricated a continuous-flow magnetic cell sorter by integrating PDMS composite stripes under the channel [62]. Soft and hard composites were realized by mixing carbonyl iron or NdFeB powders with pre-mixed PDMS at mass ratios respectively of 75 wt % and 66 wt %. The stripes (from 50 µm to 200 µm in width) were obtained by injecting the composite into a network of parallel PDMS channels using a syringe. The NdFeB stripes were magnetized using a pulsed magnetic field system producing a field of 6 T. Then, the PDMS channel of the microdevice was irreversibly bound to the embedded stripes (Figure 1c). With this method, the magnetic stripes are embedded in a PDMS matrix, limiting the possible contact of the cells with magnetic particles and providing a topography-free surface. Bead (10 µm) trapping and deviation were performed with both composites under a continuous flow rate up to 50 µL/min. They also integrated a magnetic grid under the channel and demonstrated the trapping of magnetically labeled bacteria. Similarly, Zhou et al. embedded NdFeB-PDMS stripes, 100 µm square by 35 µm thick, which were oriented perpendicular to the flow direction under the channel [63]. The micromagnets were also obtained by injecting the composite mixture into PDMS channels using a syringe. They demonstrated a localized mixing between the superparamagnetic ferrofluid and the water flow.

Composite PDMS structures can also be integrated on the channel side to trap, deflect, or separate the targets. Nagai et al. embedded composite PDMS blocks in the channel walls to trap magnetically labeled Vorticella cells and take advantage of their spontaneous elonga-

tion and contraction motion to rotate mobile structures in the channels [64]. The composite blocks were made by mixing PDMS and $Fe_3O_4$ powder (50–100 nm in diameter) with a concentration of 38 wt %. Composite PDMS structures were also recently implemented on the channel wall for droplet microfluidic. Serra et al. reported a device for the extraction, purification, and redispersion of target molecules by coupling soft composite magnets and passive fluidics [65]. Composite blocks were obtained by mixing iron particles (1 to 6 µm) with PDMS and pouring the mixture into a CycloOlefin copolymer master. Three concentrations were compared: 44, 60, and 70 wt %. By combining the magnetic function and a fluidic capacitor, they achieved extraction and purification efficiencies greater than 95% with a maximum processing speed of 17 mm/s. Zhou et al. fabricated magnetic structures by injecting PDMS composite with a syringe in a structural channel next to the fluidic channel. The fluidic and the structural channels were fabricated in a one-step soft-lithography process. They used soft [66] and hard [67] magnetic PDMS composites and investigated the mass ratio of the magnetic powder and the shape of the microstructure (semi-circular, triangular, rectangular), the channel width, and the flow rate. The schematic of the microsystem with the soft composite microstructure adjacent to the microfluidic channel is presented in Figure 1d, as well as micro-photographies of three different shapes for the composite microstructure. They studied the deviation operated on magnetic beads, performed the separation of yeast and magnetic beads at flow rates of few µL/min, and rapidly mixed ferrofluids and distilled water [46].

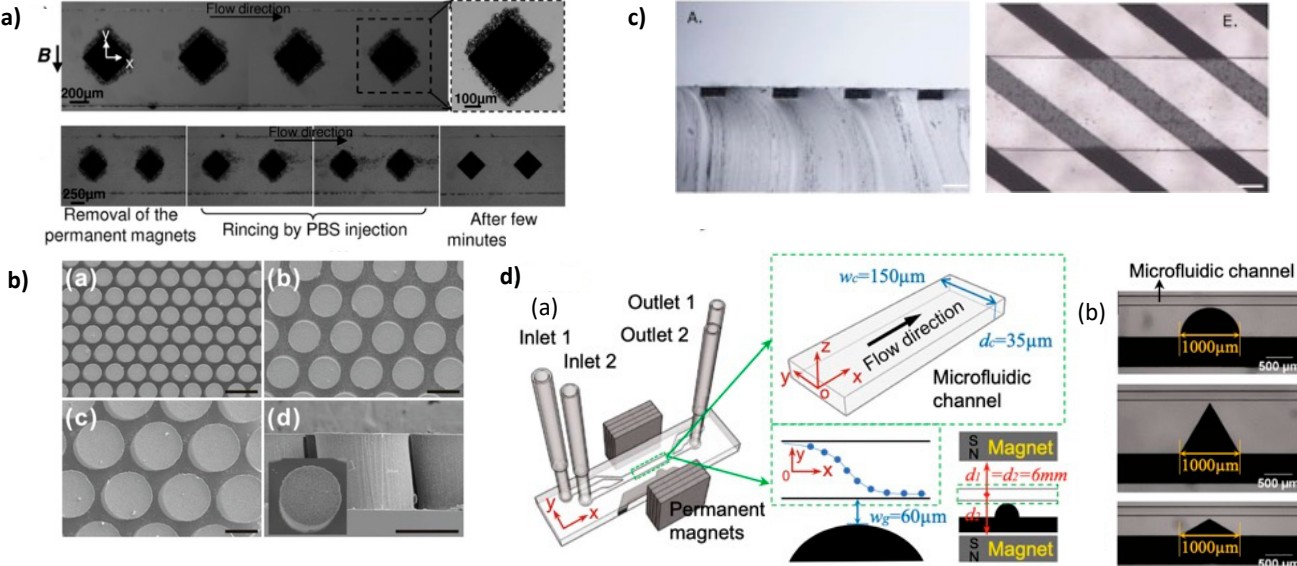

**Figure 1.** (**a**) Top: Capture of superparamagnetic beads suspended in PBS on several I-PDMS microstructures. The close-up allows distinguishing the beads trapped on a single structure. Bottom: Reversibility of the capture: the beads can be detached and collected by rinsing after removal of the magnets. Reprinted from [58], with the permission of AIP Publishing. (**b**) Scanning electron micrographs of 3D Nd-PDMS micropillar arrays of different diameters and inter-pillars distances. Scale bars: 200 µm. © IOP Publishing [60]. Reproduced with permission. All rights reserved. (**c**) Cross-sectional view of a PDMS flat slab with embedded soft magnetic stripes (A). Embedded strips on the bottom of the microfluidic chip (E). Scale bars: 100 µm. Reprinted from [62], with permission from Elsevier (**d**) (a) Schematic of the microsystem with the composite microstructure adjacent to the microfluidic channel. (b) Micro-photographies of the three different shapes for the composite microstructure next to the microfluidic channel: half circle, 60° isosceles triangle, and 120° isosceles triangle. Reprinted from [66] with permission.

## 2.2. High Concentrated PDMS Composites with Anisotropic Magnetic Properties

In the works cited above, the composites are rather highly concentrated in magnetic microparticles (concentrations greater than 30 wt %), with an isotropic dispersion of the latter in the polymer matrix. Then, the composite can be seen as a fully dense material that exhibits relatively large magnetization, even comparable with pure metallic

Ni-based alloys [81]. If any, the magnetic anisotropy is governed by the overall shape of the pattern. However, a controlled dispersion of the magnetic particles can induce a significant magnetic anisotropy contribution, which opens new prospects. The general process consists in submitting the composite mixture to a magnetic field during the polymer cross-linking step. Deman et al. self-organized an 83 wt % (38 vol%) loaded I-PDMS (iron carbonyl/PDMS) composite by applying a 130 mT magnetic field during the polymer cross-linking step [68]. In the non-reticulated polymer, the motion of the magnetic entities is essentially driven by magnetic dipolar interactions. Depending on the relative positions of two adjacent magnetized particles, the interaction can be repulsive or attractive, which leads to anisotropic mechanisms of field-induced structures such as agglomeration and self-organization [82–86]. Despite a large number of particles limiting their motion, they align in chains along the field lines, leading to a uniaxial anisotropic microstructure. Then, isotropic and anisotropic I-PDMS patterns were integrated by soft lithography in microfluidic devices to compare their magnetophoretic performances. An external magnetic field of 180 mT was supplied by two permanent magnets positioned on both sides of the channel. The authors demonstrated that the magnetic force exerted on superparamagnetic beads (12 μm in diameter) was two times larger for the device integrating the anisotropic I-PDMS at a distance of 150 μm from the composite wall. The increased magnetophoretic force was attributed to the 16% increase of the composite magnetization and to the local magnetic field gradient originating from the fine alternation of magnetic and non-magnetic regions.

Similar fabrication processes can be applied to hard magnetic materials, but additional specific approaches are to be mentioned. Chung et al. reported on the self-organization of prior magnetized NdFeB particles in a PDMS matrix in a chessboard-like multipoles pattern [69]. The alternation of up and down magnetization leads to a regular modulation of the generated stray field and high field gradients. The authors reported an original design of chip implementing multiple functions for circulating tumor cells (CTCs) isolation and profiling, including magnetic depletion obtained with the self-ordered NdFeB-PDMS composite layer on the bottom layer of the microchannel, size-selective cell capture, and on-chip molecular staining. The magnet filter depletes leukocytes and the size-sorter region traps individual cells at predefined locations. They integrated a chaotic mixer by microstructuring the top of the channel in a herringbone shape in order to deflect the cells to the magnetic layer at the bottom of the channel and enhance trapping efficiency. The enrichment ratio was enhanced by more than 30-fold with the chaotic mixture.

### 2.3. Low Concentrated PDMS Composites with Anisotropic Magnetic Properties

In contrast, when the volume fraction of the magnetic entities (particles [21,71], pillars [87,88], or fibers [89]) is reduced to few volume percentages, typically less than 10%, individual micrometer-sized magnetic flux sources can be formed and organized in regular patterns at the micrometer scale within the non-magnetic polymer matrix. To obtain well-organized arrays of micropatterns in PDMS, Le Roy et al. prepared micropillars on Si substrates topographically patterned by deep reactive ion etching and covered with 10 μm thick FeCo or NdFeB films. Then, the magnetic pillar arrays were transferred in a PDMS membrane [22]. This approach offers a high quality of replication as well as a high control over the shape and the size of the magnetic structures. In turn, it involves advanced microfabrication processes for the deposition and the micropatterning. In this section, we present the alternative offered by self-ordered low concentrated–polymer composites for magnetophoretic functions.

As described previously for high concentrated composites, low concentrated composites are submitted to a magnetic field during the polymer cross-linking step. Thus, regular magnetic patterns of micrometric size and large aspect ratio can be obtained.

#### 2.3.1. Preparation under Uniform Field

A uniform magnetic field induces a uniaxial symmetry in the dipolar interactions. These are attractive along the applied magnetic field direction but repulsive within the

normal plane with respect to the applied field direction. Therefore, a uniform field promotes the formation of 1D agglomerates in the volume of the composite. Figure 2 shows the microstructure and the magnetic properties of carbonyl iron particles/PDMS composite (I-PDMS) when prepared under a uniform magnetic field of 130 mT at different concentrations. The as-formed 1D agglomerates exhibit a uniaxial magnetic anisotropy with a higher susceptibility when magnetized along their long axis. Here, the anisotropy is assessed by the ratio between the components of the magnetic susceptibility tensor in the direction of the agglomerates' long axes and in a perpendicular direction. The magnetic characterizations of this material [90] showed that the overall array anisotropy goes through a maximum when the volume fraction of carbonyl iron particles is around 1.5%. These observations could be explained by the crossover between (i) the increased contribution of discontinuous chains; i.e., they are less anisotropic as the concentration is reduced; and (ii) the demagnetizing dipolar interactions strengthen as the density of agglomerates increases. Based on their magnetic characterizations, Le Roy et al. estimated that individual agglomerates hold a magnetization of 797 kA/m [81], which is comparable to pure metallic materials, such as nickel and nickel-based alloys.

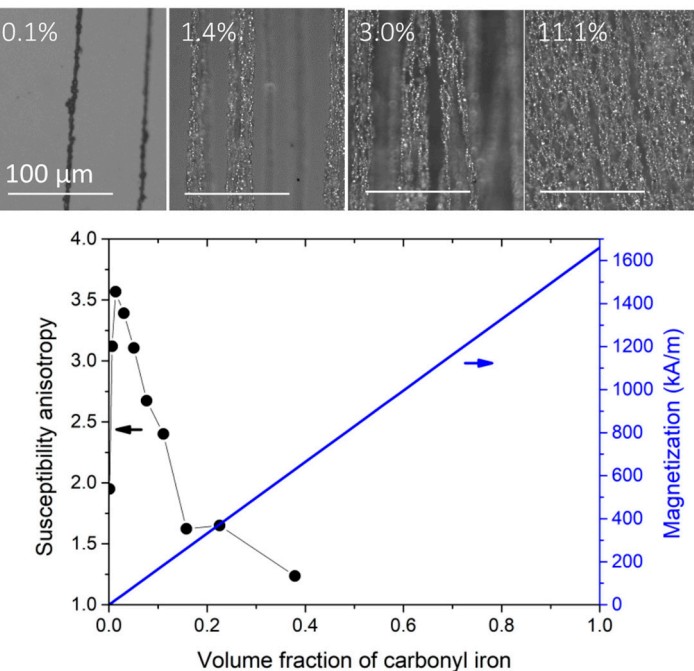

**Figure 2.** Anisotropic I-PDMS, prepared with an applied magnetic field of 130 mT. The top images are top views of I-PDMS membranes filled with volume concentrations of 0.1%, 1.4%, 3.0%, and 11.1% (from left to right). The columnar agglomerates are aligned in the direction of the applied magnetic field during the reticulation (vertical in the images). The graph shows the evolution of the anisotropy and the overall composite magnetization of the I-PDMS membranes. The susceptibility ratio is the ratio between the low field susceptibility in the direction of the applied field during the preparation and the in-plane perpendicular direction. Adapted with permission from [81] (http://creativecommons.org/licenses/by/4.0/).

An interesting feature of anisotropic I-PDMS is that it gives access, in its simplest implementation, to micro-patterns with interesting geometries (elongated patterns pointing toward the membrane's surface), which contrasts with standard micro-fabrication routes based on films. Mekkaoui et al. implemented I-PDMS, with carbonyl iron fraction of 1 and 5 wt % (0.13 and 0.68 vol %), in microfluidic devices [70] where the I-PDMS constitutes the channel's floor, exhibiting high densities of magnetic traps, of 1500 traps/mm$^2$ and 5000 traps/mm$^2$, for these two concentrations (Figure 3a). The authors assessed the magnetophoretic trapping performances on flowing superparamagnetic beads. They

measured a trapping throughput as high as 7100 trapped beads per minute at a flow rate of 0.83 µL/s and a remarkable trapping efficiency of 99.94%. Recently, Descamps et al. applied the same fabrication process with NdFeB particles instead of Fe, which can then be permanently magnetized. In the same way as for I-PDMS membranes of Mekkaoui et al., arrays of 1D agglomerates were obtained with 480 traps/mm$^2$ for a NdFeB fraction of 1 wt % [71]. These NdFeB traps were implemented for magnetophoretic trapping of superparamagnetic beads and magnetically labeled white blood cells [72].

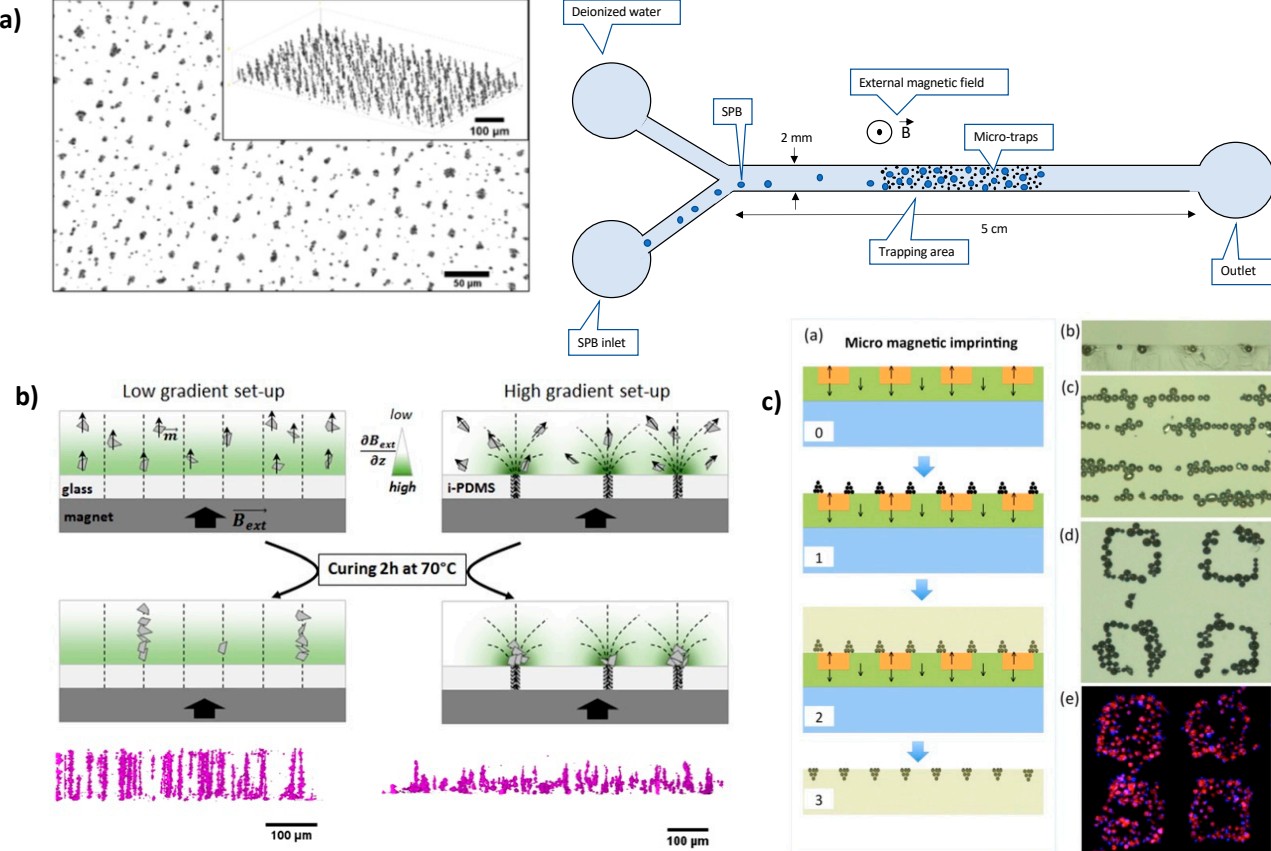

**Figure 3.** Examples of microstructure engineering by replication of a magnetic template. (**a**) X-ray tomography images of microstructured I-PDMS presenting arrays of columnar iron particle agglomerates. Adapted with permission from [70]. Copyright (2020) American Chemical Society. (**b**) Schematic views and X-ray tomography images of microstructured NdFeB@PDMS composite membranes prepared with two magnetic field patterns, leading to arrays of either columnar or conic agglomerates of NdFeB particles. Adapted with permission from [71]. (**c**) Fabrication scheme of NdFeB-PDMS membrane by micro-magnetic imprinting. Reprinted from [21] with permission of AIP Publishing.

The approach that consists in applying a nearly uniform magnetic field to a weakly concentrated PDMS composite during its curing is relatively simple to implement and is well suited for magnetic trapping of individual bio-entities, including cells that have typical sizes comparable with the trap sizes. In turn, it restricts the shape of agglomerates to 1D structures and does not allow independently tuning the lateral size and the density of traps. To form other geometries of micro-patterns, strategies of replication were developed with non-uniform magnetic field templates.

### 2.3.2. Preparation under Magnetic Field Gradient

In the same aforementioned work, Descamps et al. compared the microstructure obtained with a uniform field and a non-uniform field applied during the crosslinking step [71]. In order to submit the composite layer to magnetic field gradients, they replaced the glass slide substrate with an I-PDMS membrane structured in an array of carbonyl

iron-particle columns. The additional magnetic field template of the I-PDMS membrane led to a concentration of the NdFeB particles at the surface of the NdFeB-PDMS, forming arrays of conic agglomerates instead of columnar agglomerates (Figure 3b). The measured forces on superparamagnetic beads were increased by a factor of two, which was attributed to the improved compactness of the agglomerates. They also demonstrated that this approach offers to tune the density of microtraps as it scales with the areal density of traps in the I-PDMS template.

Earlier, Dempsey et al. used a continuous magnetic film of NdFeB [21], with written up and down domains as a master to organize NdFeB particles dispersed within non-reticulated PDMS. The method they had developed before to write domains in a permanent magnet film [91] broadens the range of pattern geometries as the NdFeB particles concentrate in the regions of magnetic field maxima, typically at the frontier between adjacent up and down domains. The resulting free-standing membrane of NdFeB-PDMS was permanently magnetized under a high field and served at remanence to trap cells (Figure 3c).

More recently, Bidan et al. applied a similar approach using a template made of topographically patterned NdFeB thick films [88], with an intercalated thin plastic foil, on which they poured a mixture of individual pillars and liquid PDMS. Doing so, the authors obtained lines of pillars regularly spaced within a low rigidity PDMS matrix that served as a magnetostrictive substrate for monitoring cell growth under controlled mechanical stress.

### 3. Magnetic Fluids

Other magnetic polymers than PDMS composites have also been introduced in microsystems to trap or sort magnetic targets. As with ferrofluids, some examples can be found with magnetically charged liquids used to sort cells. Myklatun et al. used a ferrofluid and a non-magnetic fluid flowing in two contiguous channels [73]. An external magnetic field magnetizes the ferrofluid (made of 17 vol% of 10 nm magnetite particles) and thus creates a gradient field (up to 1700 T/m) that attracts magnetic cells (macrophages containing phagocytosed magnetic nanoparticles and intrinsically magnetotactic bacteria) in the side channel (Figure 4a). Thus, the separation efficiency was 10 times increased compared to the separation with a magnet alone. Moreover, the advantage of this approach is that the parallel channels avoid direct contact between the magnetic structures and the solution containing the objects to sort. Abonnenc et al. integrated an array of 100 μm wide magnetic ink tracks perpendicular and on each side of a 100 μm wide channel surrounded by two permanent magnets in attractive configuration (with an inter-track gap of 200 μm and an inter-magnet gap of 2 mm) [74]. The microsystem was made of polyethylene-terephthalate, and the distance between the channel and the magnetic tracks was 10 μm. The magnetic ink was home-prepared and made of 1 g of carbon ink mixed with 200 mg of iron particles (10 μm). The tracks locally concentrate the magnetic flux, which produces large magnetic field gradients. The number of tracks, varying from 8 to 34 (with 2 to 10 mm long magnets respectively), improves the trapping efficiency of the superparamagnetic beads (300 nm in diameter) in the channel, which was up to a 300% increase compared to the trapping obtained with the magnets alone.

Magnetic liquids present also the advantage to easily make pre-defined magnetic structures in a microfluidic system in order to capture cells. Sun et al. filled stripe-like structures in a PDMS mold with a ferrite ink composed of cobalt ferrite nanoparticles (5–13 nm in diameter) suspended in a water solution that produces solid and controlled traps after water evaporation [75]. The ferrite ink was injected into an array of microfluidic channels fabricated by soft lithography in a PDMS mold put in contact with a glass slide. Then, the device was baked, and the PDMS mold was removed. A new microfluidic channel (960 μm wide, 100 μm thick) larger than the hard ferrite ink stripe array (200 μm wide, 4.5 μm thick, inter-stripes gap of 200 μm) was bonded to the glass substrate, thus integrating the magnetic stripes inside the channel. The authors reported the capture efficiency of immunomagnetically labeled CD11b+ cells (with beads of 4.5 μm in diameter) ranging

from 91% to 39% under flow velocities of 4 to 20 mm/s. Recently, a more complex microfluidic system combined soft iron-filled channels to act as magnetic micro-concentrators to intensify the field gradient with a series of filters specifically designed to isolate CTC cells from magnetically labeled leukocytes [76] (Figure 4b).

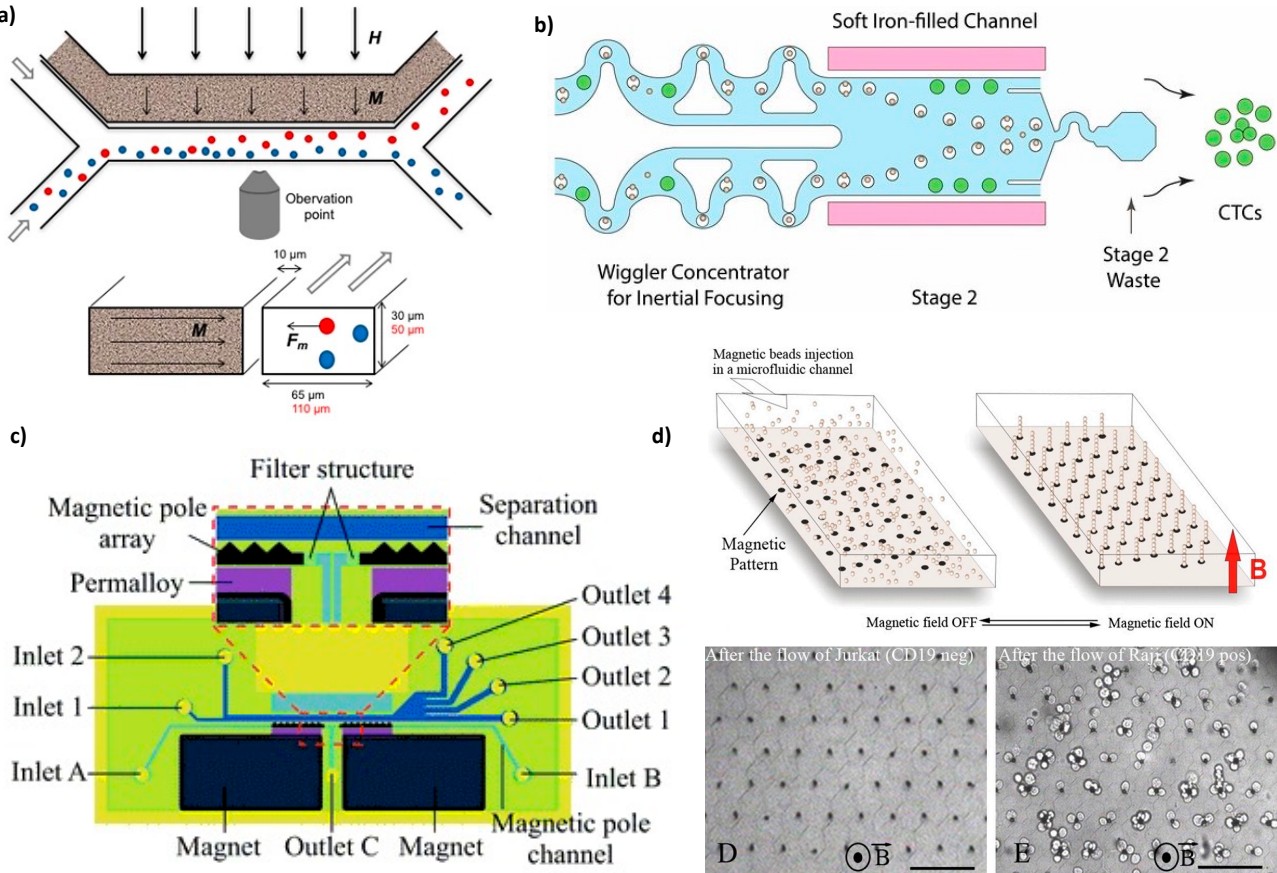

**Figure 4.** Microdevices with magnetically charged liquids used to trap or sort cells. (**a**) Design of the device seen from above (top) and as a cross-section through the channels (bottom). In the presence of an external magnetic field (thick arrows, H), the ferrofluid in the channel at 10 μm from the sorting channel magnetizes (thin arrows, M) and produces an attractive force (Fm) acting on magnetic cells (red), whereas non-magnetic cells (blue) continue in the sample flow. Reprinted from [73] with permission (http://creativecommons.org/licenses/by/4.0/). (**b**) Illustration of a magnetic sorter able to deplete ≈3 billion WBCs (white blood cells) per hour from concentrated leukapheresis products. A first size-based inertial separation removes RBCs (red blood cells) and platelets. In stage 2, two channels, one on each side of the sorting channel are compactly packed with soft magnetic iron particles. Immunomagnetically labeled WBCs (white dots with beads) are deflected in the waste well 2, whereas CTCs (circulating tumor cells, green dots) are collected at the chip output. Reprinted from [76] with permission (http://creativecommons.org/licenses/by/4.0/). (**c**) Chip detail and an enlarged view of the separation region: the microfluidic structures are made of PDMS by soft lithography and bonded to a glass substrate, the magnetic pole arrays are filled with $Fe_3O_4$ powder, and the magnets are placed in opposite directions. Reprinted from [77], with the permission of the Royal Society of Chemistry (RSC). (**d**) Design of the Ephesia system (top): beads coated with an antibody are injected in a microchannel patterned with an array of magnetic ink, and the application of an external vertical magnetic field (red arrow) induces the formation of regular bead columns on top of the ink dots. Optical micrographs showing the application of the device (bottom): columns after the passage of 1000 Jurkat cells that exited the array (left) and columns after the passage of 400 Raji cells that were captured (right). Reprinted from [78] with permission of PNAS. Scale bars: 80 μm.

　　　　Zeng et al. proposed a device where a ferrofluid containing only 0.01% of $Fe_3O_4$ nanoparticles allows the separation of a mixture of non-magnetic particles of different sizes (1 and 0.2 μm) by negative magnetophoresis [77]. The chip consists of two microchannels to fabricate magnetic pole arrays (filled with $Fe_3O_4$ powder, at 3 μm from the ca. 200 μm-wide

separation channel), two rectangular-shaped permalloys (30 μm in thickness), and two NdFeB magnets (15 mm × 15 mm × 10 mm, at 600 μm from the separation channel) (Figure 4c). Thus, a magnetic field gradient greater than $10^5$ T.m$^{-1}$ is generated, and the particles are collected in the different outlets depending on their size and the flow rate.

Finally, a different approach is to use the self-organization properties of magnetic particles to produce magnetic traps inside the microfluidic channel. J. L. Viovy's group has been a pioneer in assembling biofunctionalized superparamagnetic beads (1 μm and 0.57 μm ± 5% of diameter) in a columnar organization to separate large DNA (from 10 to 160 kilobase pairs) [92,93]. Saliba et al. adapted this approach for micrometric objects, such as eukaryotic cells, by growing the magnetic columns of functionalized superparamagnetic beads onto a magnetic pattern obtained by microcontact printing of a water-based ferrofluid (a commercial magnetic ink) onto glass [78]. The self-assembly in columns was obtained by applying a cooled electromagnet coil delivering a uniform magnetic field of 20 mT, perpendicular to the coverslip's plane. The size of the magnetic array is of cone-like dots of 5+/−1 μm of diameter and 475+/−25 nm of height, and the diameter of the beads is 4.5 μm. The obtained columns have the same height of the channel (ca. 50 μm), have a center-to-center bead spacing of 40 μm, and start to detach for average flow velocities between 800 μm/s and 1 mm/s (compared to 20 m/s for the non-templated magnetic arrays), but the template of magnetic dots remains undamaged. Moreover, the authors demonstrated the cell viability over 12 h of culture, which allows biological tests on the sample. With this approach, called "Ephesia", they obtained about 94% of capture yield (under a flow rate of a few μL/min and a range of ten to hundred cells/s). Figure 4d shows the principle and optical graphs of cells individually visible on the traps, according to the antibody functionalization of the beads.

## 4. Conclusions

The composite approach breaks with the microfabrication techniques of microelectronics and offers a promising alternative in terms of cost and simplicity of manufacture and also in terms of the integration of magnetic functions in microsystems. Here, we have reviewed the different studies using integrated magnetic polymers to manipulate micro-objects by magnetophoresis. The choice of the magnetic powder (size, nature, shape, concentration), the polymer matrix, and the microstructuring of the composite allows the design of a variety of magnetic functions. Composite PDMS is certainly the most versatile in terms of microfabrication. Soft lithography structuring allows the integration of composite structures into the channel, at its sides, or underneath. At high concentration, it can be seen as a continuous material. If it is cross-linked under a magnetic field, it can exhibit anisotropic magnetic properties, and at low concentration, it is possible to obtain regular magnetic patterns in the polymer matrix with aspect ratios that are difficult to achieve with conventional microfabrication techniques. Magnetic inks and ferrofluids have also been used to trap or sort magnetic targets into microsystems.

In conclusion, the different designs reported in this review show the variety offered by magnetic polymers in terms of micro-fabrication and magnetic functions. Thus far, PDMS remains the most commonly employed polymer base in magnetic composites. However, thermoplastic polymers, such as polymehtylmethacrylate (PMMA), polycarbonate (PC), and cyclic olefin copolymer (COC), are commonly used in microfluidics and lab-on-a-chip devices [94,95] and may offer new avenues in the spread of magnetic polymer-based devices. In addition, micro-structured magnetic polymers in microfluidic devices are mainly performed by casting and soft lithography processes, but other printing methods, including 3D printing, are currently explored in soft electronics and soft robotics [96]. These cutting-edge techniques may be investigated by the microfluidic community to implement specific applications, including cell sorting.

**Author Contributions:** Writing—original draft preparation, L.D., D.L.R., C.T., A.-l.D.; writing—review and editing, L.D., D.L.R., C.T., A.-l.D. All authors have read and agreed to the published version of the manuscript.

**Funding:** This research was funded by PACK AMBITION RECHERCHE AuRA, grant LUTON number 1701103701-40890.

**Institutional Review Board Statement:** Not applicable.

**Informed Consent Statement:** Not applicable.

**Data Availability Statement:** Not applicable.

**Acknowledgments:** L.D. acknowledges the doctoral school EEA–Univ. Lyon 1 for its support.

**Conflicts of Interest:** The authors declare no conflict of interest.

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
