# Peer review of "Magnetic Polymers for Magnetophoretic Separation in Microfluidic Devices"

_magnetochemistry, doi:10.3390/magnetochemistry7070100_

Round 1

Reviewer 1 Report

The review by Descamps and co-authors discusses state of the art of research works implementing magnetic polymers to trap or sort magnetic micro-beads or magnetically labeled cells in microfluidic devices. The authors call these materials as “magnetic polymer”, but they are polymer molecules ‘doped’ with magnetic nanoparticles.

Several aspects are described in the review, which is focused mainly on the polydimethylsiloxane (PDMS) elastomers: high-concentrated PDMS solutions; the peculiarities brought by anisotropic magnetic particles; preparation under uniform or gradient external magnetic field. Some examples of trapping and/or sorting of magnetic targets with the help of magnetic inks (ferrofluids) are also discussed.

On my opinion, the review deserves its publication in Magnetochemistry.

Author Response

We thank the reviewer for his (her) positive evaluation regarding our review.

Reviewer 2 Report

This manuscript is a review on the use of magnetic polymers in microfluidic devices for magnetic separation. The review is in general well written and cites most of the relevant articles on the considered topic. A few other papers can be included to the reference list, as specified below. I recommend this paper for publication in Magnetochemistry after minor revisions of the following points:

  1. 1 on the bottom: “The magnetic force can be positive or negative”. It would be better to explain that “positive” force or “positive” magnetophoresis are referred to the magnetic force oriented along the magnetic field gradient, while “negative” magnetic force or “negative” magnetophoresis are referred to the force oriented to the direction opposite to the field gradient.
  2. 8, 1st paragraph: “Here the anisotropy is assessed by the low-field susceptibility ratio…” The term “low-field susceptibility ratio” is clearly defined in the caption of Figure 2. Please define it in the text. The word “low” may appear confusing because at fast reading it may be related to “low magnetic anisotropy” of field-induced structures, while the authors certainly mean high anisotropy of the magnetic susceptibility (high difference between the longitudinal and the transverse components of the susceptibility tensor) under low-field limit. I suggest rephrasing this sentence to render its meaning less confusing.
  3. It could be helpful to define the term “magnetic ink”. Is this term appropriate for all the systems described in Section 3, especially those shown on Figure 4a and 4b?
  4. Please provide a more general reference on the phenomenon of magnetophoresis in addition to (or replacing) the references 1-3.
  5. The Review could be further enriched by the following references:
  • A reference on a seminal book on magnetic cell separation: M. Zborowski and J. J. Chalmers, Magnetic Cell Separation (Elsevier, Amsterdam, 2008).
  • A reference on an exhaustive review on magneto-microfluidic technics used for immunoassays: M. A. M. Gijs, F. Lacharme, and U. Lehmann, Chem. Rev. 110, 1518 (2010).
  • It could be useful to mention possible enhancement of magnetophoresis by magnetic particle aggregation under magnetic field and/or attractive colloidal interparticle forces - the phenomenon which often occurs when magnetic particles (SPIONs, magnetic beads, magnetic liposomes, …) are dispersed in physiological media. In this context, reviews by Leong et al. Langmuir 36 (2020) 8033-8055 on cooperative magnetophoresis and by Kuzhir et al. J. Magn. Magn. Mater. 431 (2017) 84-90 on magnetic separation of nanoparticles enhanced by their field-induced aggregation could be mentioned.
  • While reviewing magnetic arrays fabricated by electroplating and embedded into microfluidic channels (references 20-22 in Introduction), the works of T. Deng, M. Prentiss, and G. M. Whitesides, Appl. Phys. Lett. 80, 461 (2002) on RBC separation on magnetized micropillars and Orlandi et al. Phys. Rev. E 93 (2016) 062604 on magnetic nanoparticle separation on nickel micropillar arrays could be helpful.
  • Another example of I-PDMS system is a microfluidic channel with embedded iron containing PDMS micropillar fabricated in one step with iron free PDMS substrate and used for separation of magnetic nanoparticles: Ezzaier et al. Nanomaterials 8(2018) 623.
  • While describing field-induced structuring in magnetic polymers, the authors are invited to refer to exhaustive literature on magnetorheological elastomers or ferrogels. This literature describes in a detailed manner the physical aspects of the structuring, such as magnetic anisotropy of the microstructure, spatial period of the magnetic columns, kinetics of structuring under field as compared to kinetics of polymer cross-linking; possible reversible or irreversible structural changes in magnetic polymers after cross-linking under the action of applied magnetic field, and so on.

Reviewer 3 Report

In this manuscript, the authors reviewed the state of the art of research works of magnetic polymers to trap or sort magnetic micro-beads in microfluidic devices.

I should say this is a well-written review paper. I personally enjoyed reading this manuscript and learned from it.

The introduction is well developed, the body of the manuscript is well written and organized, finally, the conclusion is reasonable.

Congratulations 

Author Response

We sincerely thank the reviewer for his (her) positive comments and its favorable opinion.